# Unleashing Region Understanding in Intermediate Layers for MLLM-based Referring Expression Generation

**Yaoyuan Liang**[1]*, **Zhuojun Cai**[1]*, **Jian Xu**[1], **Guanbo Huang**[1], **Yiran Wang**[1], **Xiao Liang**[1],
**Jiahao Liu**[2], **Ziran Li**[2], **Jingang Wang**[2], **Shao-Lun Huang**[1]†
[1]Tsinghua Shenzhen International Graduate School, Tsinghua University
[2]Meituan Inc.

## Abstract

The Multi-modal Large Language Model (MLLM) based Referring Expression Generation (REG) task has gained increasing popularity, which aims to generate an unambiguous text description that applies to exactly one object or region in the image by leveraging foundation models. We empirically found that there exists a potential trade-off between the detailedness and the correctness of the descriptions for the referring objects. On the one hand, generating sentences with more details is usually required in order to provide more precise object descriptions. On the other hand, complicated sentences could easily increase the probability of hallucinations. To address this issue, we propose a training-free framework, named as "unleash-then-eliminate", which first elicits the latent information in the intermediate layers, and then adopts a cycle-consistency-based decoding method to alleviate the production of hallucinations. Furthermore, to reduce the computational load of cycle-consistency-based decoding, we devise a Probing-based Importance Estimation method to statistically estimate the importance weights of intermediate layers within a subset. These importance weights are then incorporated into the decoding process over the entire dataset, intervening in the next token prediction from intermediate layers. Extensive experiments conducted on the RefCOCOg and PHD benchmarks show that our proposed framework could outperform existing methods on both semantic and hallucination-related metrics. Code will be made available in `https://github.com/Glupayy/unleash-eliminate`.

## 1 Introduction

Referring expression generation (REG) [11, 29, 31, 53, 54] is a task to generate an unambiguous text description that applies to exactly one appointed object or region in the image. A good expression should be distinguishable enough to ensure that the listener can identify the unique target among various objects within the same image. With the great success achieved by large language models (LLMs), multi-modal large language models (MLLMs) [1, 9, 12, 27, 56, 61] have been introduced to perform this task and become increasingly popular in the research community. Some representative works [6, 37, 52, 55, 57] conduct visual instruction tuning on specialized region-involved multi-modal corpus and successfully empower MLLMs with the ability of region-level understanding.

Though certain progress has been made, MLLMs themselves suffer from object hallucination [14, 23, 46, 50, 60]. Research suggests that the mechanism behind MLLMs' hallucinations could be related to the over-reliance on prior knowledge of LLMs rather than the multi-modal context provided by

---

*Equal contribution.
†Corresponding author.

38th Conference on Neural Information Processing Systems (NeurIPS 2024).

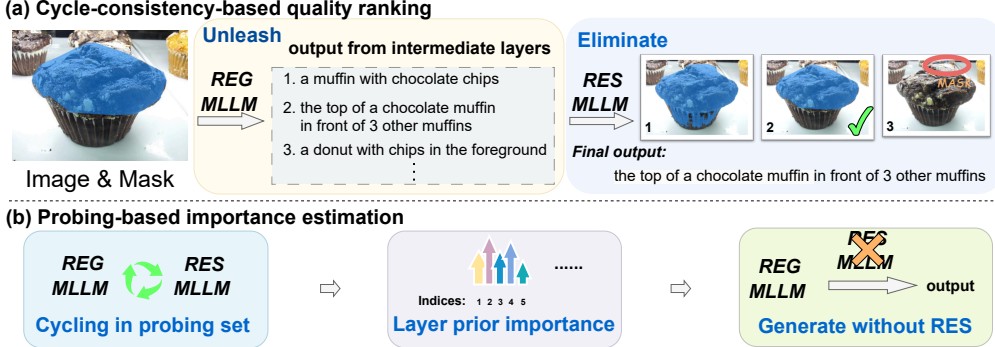

Figure 1: Illustration of our method. (a) Given an image and a mask appointed to the target object, we first unleash the descriptive regional information of the middle layers and gain various candidate captions. These outputs are sent to a RES model that serves as a "listener" and the "listener" eliminates the inaccurate candidates. (b) Our proposed strategy to diminish the computational load of RES. First estimate the layer prior importance on a probe set with RES, then leverage it for the RES-free next token prediction.

the input [14, 19, 50]. Therefore, MLLM-based REG models naturally inherit the aforementioned hallucination issues. Additionally, a further complication arises in the context of region-level understanding, where models are required to generate precise and identifiable descriptions for specific regions. To achieve that, the MLLMs sometimes have to make use of the surroundings for reference while effectively avoiding out-of-region information distorting the description of the target object, e.g. incorrectly attributing characteristics of other objects to the targeted region (Figure 7 in appendix). This requirement exacerbates the issue of attribute-level hallucinations [3, 50]. In this paper, we aim to explore this potential trade-off between the detailed description and accurate targeting of referring objects in MLLM-based REG task. To be specific, providing precise object descriptions necessitates generating sentences with more details, which results in longer sentences. Meanwhile, the text with increased granularity (length) is more likely to contain inaccurate or spurious information, commonly defined as "hallucinations." Table 1 shows an example of quantitative analysis.

To address the trade-off between information richness and reliability, we propose a novel approach called "unleash-then-eliminate," as depicted in Figure 1. We observed that the alignment of region-level multi-modal information does not maintain a monotonic progression during inter-layer transitions. Moreover, when multi-modal hidden states are projected into the language space using the language head, the intermediate layers sometimes hold more descriptive region information than the final layer. These observations (Section 3.2) imply that the most suitable layer for each referred region should not be solely confined to the final layer. Accordingly, we adopt contrastive decoding [8, 22] to unleash the object information contained in the intermediate layers. To eliminate unsuitable candidate outputs, we leverage the dual task of REG: Referring expression segmentation (RES) [37, 48], which aims to segment a target object mask from the entire image given a sentence describing the object. Ideally, for the same object, the region input (formed as a mask) for the REG task and the mask output from the RES task should exhibit cycle consistency. Based on this insight, we propose a cycle-consistency-based decoding method, which enables us to choose among multiple outputs based on their descriptive quality thus reducing hallucinations while maintaining the richness of the output sentences. Furthermore, considering the need to diminish the computational load of RES, we develop a hybrid layer importance measurement strategy to select the best layer for each token during the next word prediction. This strategy leverages both the layer-wise prior importance estimated over probing subset, and the Jensen-Shannon divergence [8] between the logits of each candidate layer and the last layer. With the layer-wise prior importance, the MLLM-based REG model maintains promising performance in reducing hallucinations and enhancing the granularity of the generated sentences, even without the assistance of RES model. Extensive experiments conducted on the RefCOCOg [33] and PHD [28] benchmark demonstrate that our proposed framework surpasses existing methods on both semantic and hallucination-related metrics.

## 2 Related works

**Region-level understanding in Multimodal Large Language Models.** Significant progress has been made in unleashing the region-level understanding ability in MLLMs [6, 18, 37, 52, 55, 57]. To

incorporate region-level information into sequence generation of MLLMs, some approaches [6, 52] integrate bounding box coordinates into the language input in the form of natural language prompts. Ferret [52] proposes a spatial-aware visual sampler that enables the arbitrary shapes of visual prompts. The current proposed Osprey [55] unlocked the capability of pixel understanding, alleviating the influence of irrelevant information in the visual prompt inputs. These methods have propelled the REG task into the era of MLLMs. Considering that MLLMs incorporate extensive knowledge from unimodal and multimodal pre-trained corpora, there is a potential for the REG task to leverage this inherent knowledge within the models to generate more specific and detailed expressions without additional training, thus more effectively addressing real-world applications [21, 39, 41].

**Decoding strategies to mitigate hallucination in Large Language Models.** Large language models are pre-trained on unlabeled corpora to acquire extensive world knowledge and subsequently undergo post-training to learn to follow instructions [34] and align with human preferences [2]. This systematic pre- and post-training pipeline makes them powerful at solving a wide range of NLP tasks [16, 36, 40, 43]. However, some studies indicate that they may fail to accurately assess their own knowledge [51] and often exhibit overconfidence in their responses [47], which results in hallucinations [59]. To mitigate these issues, recent research [5, 8, 20] proposes inference-time decoding strategies for trained LLMs to find latent knowledge inside the internal activations without additional training. C. Burns *et al.* [5] introduce a Contrast-Consistent Search (CCS) algorithm to identify a direction of truth in the activation space of LLMs that remains consistent across negations, thereby reducing generated errors. Based on the discovery of CCS, ITI [20] dives deep into attention heads and suggests shifting model activations alongside factuality-related heads during inference to help reduce hallucinations. Besides "finding the direction of truth," DoLa [8] proposes contrastive decoding by comparing the differences in logits between the projections of later and earlier layers to better surface factual knowledge and reduce the generation of incorrect facts. In line with the motivations of [5, 20], our investigation uncovers that well-trained MLLMs' intermediate layers differ in multi-modal alignment and region-level understanding capabilities. These observations inspire us to devise an inference-time decoding strategy that combines latent knowledge from multiple layers (with prior importance) to alleviate hallucinations.

**Hallucination in Multi-modal Large Language Models.** In the realm of MLLMs, "hallucination" typically refers to "object hallucination," where the models generate plausible outputs containing objects that are either absent from or mismatched with the images [15, 19, 24, 38, 50], and is commonly categorized into three types: category, attribute and relation hallucinations [3, 49]. Some efforts based on instruction tuning have been made to mitigate this issue in MLLMs. LRV-Instruction [26] introduces a dataset with positive instructions and unique negative prompts with different semantic levels to better align responses with image content. HACL [15] explores the vision-language embedding space, using contrastive learning to separate non-hallucinated from hallucinated texts. Without instruction tuning, Woodpecker [50] offers a training-free pipeline for hallucination correction, using expert models to enrich image context and ensuring each phase is interpretable by a step-by-step correction process. In another line of work, some efforts have sought new decoding strategies to avoid relying on extensive additional data and training: Opera [14] tackles the partial over-trust issue in decoding by applying a penalty to the model logits during beam-search decoding. VCD [19] links object hallucinations to biases and language priors, contrasting outputs from distorted and original visuals to ensure consistent generation. Compared to existing studies, our method initially enriches region-level context from intermediate layers rather than external expert models or knowledge base and provides a multi-layer ensembling solution to mitigate hallucinations.

## 3 Method

### 3.1 Preliminary

Leveraging the nuanced language representation capabilities inherent in multi-modal large language models, recent investigations have unlocked the REG [6, 37, 52, 55] and RES [18, 37] capabilities of MLLMs. To maintain clarity and conciseness, we have omitted the discussion on instruction tuning and assume that the described tasks are well-instructed by specific prompts. In this work, we utilize a "mask" as the region prompt.

**Referring expression generation.** The MLLM-based REG model is typically structured around three principal components: a visual encoder (e.g., CLIP [35] with a linear adapter), a region-encoder,

and an LLM base (e.g., LLaMA [42] or Vicuna [7]). Given an image $I$, the visual encoder first extracts the visual information, then the linear adapter projects it into $I_f \in \mathbb{R}^{HW \times C}$, where $HW$ denotes the flattened visual token length and $C$ denotes the hidden state dimension of LLM base. A referred region $M$ (presented as a mask) is encoded as $M_f \in \mathbb{R}^{Q \times C}$ by region encoder (e.g., a CNN-based extractor [55] or an ROI pooling layer [37, 57]), where $Q$ denotes the length of encoded region prompt tokens. Taking the visual feature $I_f$, regional feature $M_f$, and an embedded instruction text $X_f \in \mathbb{R}^{L \times C}$ as input, where $L$ represents the length of the instruction in tokens, the model concatenates them and forms a multi-modal input sequence $S \in \mathbb{R}^{(HW+Q+L) \times C}$.

Regarding the core architecture of the LLM base, the multi-modal sequence $S$ is successively processed by the $N$ stacked transformer layers. Eventually, an affine layer $\phi(\cdot)$ serves as a language model head to predict the probability of the next token $y_t$ over the vocabulary set $\mathcal{V}$. The logits for the token $y_t$, given the sequence $S$ and all preceding tokens $y_{<t}$, are computed as follows:

$$\text{logit}_N(y_t|S, y_{<t}) = \phi(h_t^{(N)}), \quad y_t \in \mathcal{V}, \tag{1}$$

where $h_t^{(N)}$ denotes the hidden state of the last transformer layer. The probability of the next token $y_t$ is then given by:

$$p(y_t|S, y_{<t}) = \text{softmax}\big(\text{logit}_N(y_t|S, y_{<t})\big), \quad y_t \in \mathcal{V}. \tag{2}$$

Through this process, the model autoregressively generates the output text $Y$ as the region description. For simplicity, we use $p(y_t|y_{<t})$ to represent $p(y_t|S, y_{<t})$, and $\text{logit}_N(y_t|y_{<t})$ to represent $\text{logit}_N(y_t|S, y_{<t})$ in the following.

**Referring expression segmentation.** Recent advances in MLLM-based RES models [18, 37] largely employ a similar MLLM architecture (e.g., LLaVA [27]) as used in the REG models, with the distinction that its input contains a description targeted at a specific region, while its output is a mask that can cover the described object.

A widely adopted strategy [17, 18, 37] incorporates a `[SEG]` token, enabling the model to identify the `[SEG]` token in the output sequence as a cue for the presence of a segmentation target. A specialized MLP head $\psi$ processes the output embedding of the `[SEG]` token $h_{seg} \in \mathbb{R}^{1 \times C}$, mapping it into the prompt space of the segmentation fundamental model (e.g., SAM [17]), represented as $\tilde{h}_{seg} = \psi(h_{seg})$. The segmentation model then decodes the target mask $M_s$ from the query token $\tilde{h}_{seg}$ and provides its confidence score $CF$.

### 3.2 The intermediate layer contains descriptive information

This section focuses on our observations of intermediate layers, attempting to uncover the latent descriptive information. We adopted a region understanding model Osprey-7b [55] with $N = 32$ layers as REG model, and GLaMM [37] as RES model by default.

**Each layer has different generation tendency.** To reveal the latent information, we adopt the early-exit strategy on the REG model to generate a series of output sequences by applying the language model head (an affine layer) to the hidden states of each layer. As illustrated in Figure 2, the sequences from the early layers (layers 1 to 10) manifest as nonsensical strings of characters, indicating the suboptimal alignment between the hidden representation of shallow layers and the ultimate vocabulary space of MLLM. Interestingly, among the middle layers (about layers 20 to 25), the model begins to output sentences with semantic meaning and gradually some region-specific expressions appear. Compared with the output of the last layer, these sentences with richer semantic information can sometimes provide more discriminative descriptions for the referred object, e.g. indicating the related position between the target object and others, and/or containing high granularity of attributes. The uniqueness and descriptiveness of expressions given by the middle layers are strongly related to the objective of REG, thus the latent features of intermediate layers have notable potential to enhance the unambiguity in MLLM-based REG.

**Potential of intermediate layers.** The above observations demonstrate that different layers' understanding of the region context varies. We attempted to visualize the region-aware comprehension capabilities among the layers through two approaches. (a) With the assistance of the RES model: we randomly extracted $K = 2000$ samples from the RefCOCOg training set to form the triplets $(\mathbf{I}, \mathbf{M}, \mathbf{Y})$, which corresponds to the entire images $\mathbf{I}$, the target regions represented by masks $\mathbf{M}$, and the descriptions of regions $\mathbf{Y}$. $(\mathbf{I}, \mathbf{Y})$ are sent into an MLLM-based RES model, to harvest the projected features of `[SEG]` token $\tilde{h}_{seg}$. Subsequently, $(\mathbf{I}, \mathbf{M})$ are input into the REG model, and the

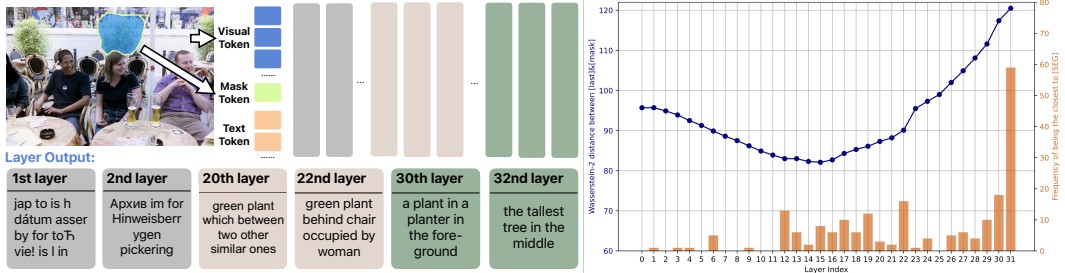

Figure 2: Different layers' understanding of the region context varies, where early layers generate rubbish, middle layers tend to generate descriptive text with higher granularity, and the final layers tend to predict shorter and more precise text. The right part shows the frequencies with which the hidden state of each layer had the smallest Wasserstein-2 distance to the [SEG] token (in orange), and the inter-layer transitions of region-level multi-modal alignment (in blue).

hidden states of the last token of each layer $\{h_i\}_{i=1}^{N}$ are obtained. Inspired by FID metric [13] and recent investigations on the latent space communication [30, 32], we first performed PCA dimensionality reduction to project both features into the same dimension, then treat the dimension-reduced [SEG] features as the anchor to calculate the Wasserstein-2 distances [13, 44, 45] between [SEG] and each intermediate hidden states. This allowed us to estimate the relative region understanding ability of each layer for the given multi-modal context of a certain object. We then calculate the frequency with which each layer had the smallest Wasserstein-2 distance to the [SEG] token. As depicted in the right of Figure 2 (orange), except for the final layer, the intermediate layers also have the potential to contain better latent information for a more precise region-related description. (b) Solely within the REG model: we investigated the multi-modal alignment process in region-level context across intermediate layers by calculating the Wasserstein-2 distance between each layer's region-encoded token [mask] and the last language token. Results are reported in Figure 2 (blue). These analyses within a well-trained MLLM show that the tokens of different modalities do not approach monotonically during inter-layer transitions. Hence we should give more chances to the intermediate layers for better region-level understanding. Detailed information can be found in the Appendix C.

## 3.3 Unleash then eliminate decoding strategy

To enhance the granularity of region-level descriptions without introducing excessive hallucinations, we propose a method that integrates contrastive decoding with cycle consistency-based ranking to screen out appropriate descriptions for the interested regions (Section 3.3.1 and 3.3.2). This approach enables us to specifically leverage the commonly overlooked latent information contained in intermediate layers and ensures the identifiable description through caption quality estimation. Under the concerns of computation efficiency, we further develop a decoding strategy to reduce the operations of cycle ranking through hybrid layer importance measurement (Section 3.3.3). It involves two kinds of layer importance calculation to influence the selection probability of candidate layers during each word prediction step. The harvest prior importance weights can be directly applied to the decoding process of the original MLLM, mitigating the hallucinations. The following subsections provide further details.

### 3.3.1 Unleash intermediate information by contrastive decoding

As depicted in Figure 2, the manifestation of region-aware information differs across layers. To underscore descriptions related to specific regions—evident in the intermediate layers but faded in the final layer, we adopt a contrastive decoding approach [8, 22] by subtracting the log probabilities of the next token produced by the intermediate layer from those of the final layer. The resulting distribution is defined as the contrastive decoding logits of specific subtractor layers. These logits instead of originally the logits from the final layer are used for generating the subsequent token. Concretely, given a set of candidate layer indices $J = \{1, \ldots, n\}$, the probability of token $y_t$ for layer $j \in J$ is:

$$p_{con_j}(y_t|y_{<t}) = \begin{cases} \text{softmax}\big(\text{logit}_N(y_t|y_{<t}) - \text{logit}_j(y_t|y_{<t})\big) & \text{if } y_t \in \mathcal{V}_{head}, \\ 0 & \text{otherwise,} \end{cases} \quad (3)$$

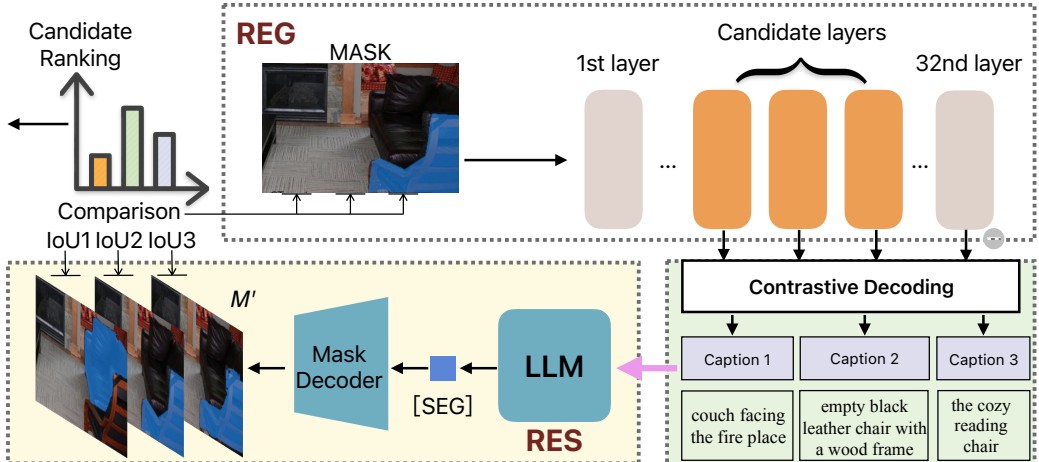

Figure 3: First, we conduct contrastive decoding by subtracting the log probabilities produced by the intermediate layer from those of the final layer. Then, each of the intermediate layers in the candidate subset suggests a probability for predicting the next token. Finally, RES model is used to estimate the performance of the captions by calculating the Intersection over Union (IoU) between MASK and $M'_j$, ranking these candidates effectively.

where $\text{logit}_j(y_t|y_{<t}) = \phi(h_t^{(j)})$, $\mathcal{V}_{head}(y_t) = \{y_i \in \mathcal{V} : p(y_i|y_{<t}) \geq \alpha \max_{w \in \mathcal{V}} p(w|y_{<t})\}$, $\alpha \in [0,1]$ is a cutoff hyperparameter that truncates the next token distribution $p(y_t|y_{<t})$ of the final layer. Following the previous works [8, 22], we set $\alpha = 0.1$ in the implementation.

After contrastive decoding, each of the intermediate layers in the candidate set suggests a probability for predicting the next token $y_t$. Finally, we harvest a sentence set $A = \{a_1, a_2, ..., a_n\}$ whose size equals to the size of candidate layers.

### 3.3.2 Cycle-consistency-based intermediate sentence quality ranking

Consider the formed triplet ($\mathbf{I}$, $\mathbf{M}$, $\mathbf{Y}$) in Section 3.2, where $\mathbf{I}$ serves as the global contextual background of a region, $\mathbf{M}$ and $\mathbf{Y}$ respectively represent two modalities of the same object. For a pair of ideal REG and RES models, it is anticipated that $\mathbf{M}$ and $\mathbf{Y}$ can be interconverted losslessly during a cyclic operation of these two models. This implies that, due to the cycle consistency between the two modalities of the same object, feeding the output of the REG model into the RES model should yield a mask consistent with the input to the REG. If the output generated from the candidate layer is overly ambiguous or polluted by hallucinations, the RES model may struggle with accurately locating the target object against the background. Based on these assumptions and observations, we utilize RES model [37] to estimate the region understanding performance of the captions generated by the candidate layers, allowing us to rank these candidates effectively.

Figure 3 illustrates the pipeline of cycle-consistency-based quality ranking. The input image $I$ and the referred region (represented as a mask) $M$ are processed by the REG model, which continuously extracts and aligns features across successive layers for the multi-modal context. After the information elicitation of intermediate layers, we harvest a set of sequence $A = \{a_1, a_2, ..., a_n\}$. Each sequence in $A$, paired with the image $I$, forms an input pair $(I, a_j)$ to feed into the RES model. The RES model then segments out the corresponding mask $M'_j$ for each input pair.

We evaluate the quality of each layer's sentences by calculating the Intersection over Union (IoU) between $M$ and $M'_j$:

$$\text{Score}_j = CF_j \cdot \frac{|M \cap M'_j|}{|M \cup M'_j|}, \tag{4}$$

where $CF_j$ refers to the output of IoU score head within the segmentation foundation model (e.g. SAM [17]), which can be interpreted as a confidence score of the generated mask. Subsequently, the candidate sentences from each layer are ranked by the $\text{Score}_j$. For each sample, the $a_j$ with the highest $\text{Score}_j$ is selected as the final sentence, and its layer is deemed the best candidate layer.

---

**Algorithm 1** Layer Prior Importance Calculation

---

1: **Input: Score** $\in \mathbb{R}^{m \times n}$        $\triangleright$ $m$: Number of subset samples; $n$: Number of candidate layers
2: **Output:** $\boldsymbol{q} \in \mathbb{R}^n, \sum_{j=1}^n q_j = 1$        $\triangleright$ Importance weight of each candidate layer
3: Initialize **count** $\leftarrow \boldsymbol{0} \in \mathbb{R}^n$        $\triangleright$ counts of times each layer has the maximum score
4: **for** $j \leftarrow 1$ **to** $n$ **do**
5:      $count_j = \sum_{i=1}^m \mathbb{1}\{j = \arg\max_{j \in J} \text{Score}_{i,j}\}$
6:      $q_j \leftarrow \frac{count_j}{m}$
7: **return** $\boldsymbol{q}$

---

### 3.3.3 Hybrid layer importance measurement

Although our cycle-consistency-based candidate ranking process improves the generation quality, it introduces additional computational load from the RES model compared to the original decoding method, significantly affecting the per-sample decoding speed. To alleviate this issue, we propose a simple yet effective strategy called *Probing-based Importance Estimation* to speed up the decoding process. This strategy involves frequency counting of each candidate layer being the optimal layer based on their performance within a probing subset. With a subset containing $m$ samples, the specifics of this calculation are outlined at Algorithm 1. The estimated weight $\boldsymbol{q}$ is then served as prior knowledge that reflects the candidate layers' importance and is then utilized over the entire dataset in the decoding process, intervening in the next-word prediction.

Furthermore, inspired by the success of distance-guided layer selection in LLMs [8], we also apply this metric as a second guidance. Concretely, we first calculate the distance between the next-token probability of the final layer and each candidate layer at the current decoding step, then the calculated values are normalized across candidate layers as follows,

$$d_j = \frac{D_j}{\sum_{i=1}^n D_i}, \tag{5}$$

where $D_j = \text{JSD}(p || p_{con_j})$ denotes the J-S divergence between the next-token probability of the final layer $p$ and the layer $p_{con_j}$. Finally, the hybrid layer importance is obtained by adding up the probing-based prior and sample-wise distance followed by a softmax normalization:

$$\tilde{\boldsymbol{q}} = \text{softmax}(\boldsymbol{d} + \boldsymbol{q}), \tag{6}$$

where $\boldsymbol{d}$ is a vector of normalized divergence values across the candidate layers. We then sample from the probability distribution $\tilde{\boldsymbol{q}}$ to select one layer among candidates to predict the next token at each decoding step, till the generation finishes.

Overall, this decoding approach first estimates a prior layer-wise importance weight from a small subset and then applies this distribution to contrastive decoding, effectively improving decoding efficiency while preserving the ability to mitigate hallucinations. In addition, our experiments (Table 3) also show that the prior importance $\boldsymbol{q}$ calculated from one (probing) dataset could be directly transferred to another dataset with similar image domain for reducing the hallucinations. This strong transferability further illuminates a new promising application scenario where the historic estimate could serve as a cold-start for inference in new environments when the probing dataset is not available.

## 4 Experiments

### 4.1 Datasets and metrics

**RefCOCOg.** The RefCOCOg [33] dataset is a classical benchmark for referring expression generation, which contains 85,474 expressions for 54,822 objects in 26,711 images. Expressions in RefCOCOg are annotated on Amazon Mechanical Turk in a non-competitive way and tend to be longer (8.43 words per sentence on average) and more expressive than RefCOCO [53] and RefCOCO+ [53]. In the MLLM-empowered REG task, the relatively short ground truth cannot fully cover the expression space of the sentences generated by the MLLM at evaluation. Therefore, our experiments focus on the METEOR metric as it is more comprehensive and flexible, allowing for a more nuanced recognition of linguistic variations.

Table 1: Comparison of generation and hallucination performance on RefCOCOg. $t$ denotes the temperature parameter. "1/8" denotes using 1/8 samples of the total dataset to estimate the layer prior importance. "full-R" denotes quality ranking on the whole dataset.

| Model | METEOR↑ | CHAIR$_S$ ↓ | CHAIR$_I$ ↓ | Recall↑ | Len | nCHAIR$_S$ ↓ | nCHAIR$_I$ ↓ |
|---|---|---|---|---|---|---|---|
| Osprey-7b ($t$=0.2) | 162.0 | **23.41** | **20.81** | 0.7631 | 7.15 | 3.2741 | 2.9105 |
| Osprey-7b ($t$=0.9) | 140.0 | 27.90 | 24.12 | 0.7514 | 8.11 | 3.4402 | 2.9741 |
| DoLa | 168.0 | 43.44 | 31.78 | 0.8196 | 23.07 | 1.8830 | 1.3775 |
| Ours (1/8) | 172.0 | 42.25 | 30.95 | 0.8211 | 22.96 | 1.8406 | 1.3484 |
| Ours (full-R) | **173.0** | 42.40 | 31.20 | **0.8237** | 23.16 | **1.8307** | **1.3472** |

**CHAIR Evaluation on Hallucinations**. The Caption Hallucination Assessment with Image Relevance (CHAIR) metric [38] is commonly used to evaluate object hallucinations that occur in image description tasks. It comprises two distinct assessment dimensions, including CHAIR$_S$ that calculates on sentence-level and CHAIR$_I$ that calculates on a more granular object-level. We observe that CHAIR only counts hallucinated objects for each "central object", which means that if the model wants to enrich the semantic information and generate new tokens, it will face the risk of increasing hallucinations compared to the shorter-sentence models. This undermines the comparability of this metric across sentences of varying lengths. Hence we propose normalized CHAIR (nCHAIR) based on CHAIR to conduct a fair comparison between sentences of different lengths. The calculation of nCHAIR$_S$ is specified as the following formula, with nCHAIR$_I$ calculated similarly:

$$\text{nCHAIR}_S = \frac{|\{\text{hallucinated sentences}\}|}{|\{\text{average number of tokens per sentence}\}| \times |\{\text{all sentences}\}|}. \tag{7}$$

**Prompted Visual Hallucination Evaluation Benchmark (PHD)**. This benchmark [28] focuses on the four major types of hallucination faced by MLLMs, namely Object hallucination, Attribute hallucination, Multi-modal conflicting hallucination, and Counter-common-sense hallucination. It evaluates and explores the hallucinations through comprehensive prompt-based tasks, which also helps identify the causes of these hallucinations. While this benchmark does not explicitly evaluate region-level hallucinations, its detailed evaluating strategies for object attribute and position are closely related to region-level understanding, allowing it to effectively indicate regional-level hallucinations.

## 4.2 Main result on referring expression generation

**Baseline region-level MLLM model and decoding method.** Table 1 presents the performance comparison of semantic quality and hallucination evaluation on RefCOCOg dataset, with METEOR and CHAIR metrics. Recent research [58] has revealed that the temperature parameter $t$ has a notable effect on the hallucination of generated sentences. In light of this, we included an analysis of the baseline region-level MLLM model, Osprey-7b [55], performing at both lower ($t = 0.2$) and higher ($t = 0.9$) temperature settings. Meanwhile, we also compared a baseline decoding method, DoLa [8], which demonstrated that leveraging contrastive decoding in vanilla LLMs can enhance the authenticity of the generated results. In comparison to the Osprey-7b [1] , the sentences generated by DoLa [8] demonstrated better performance on METEOR and nCHAIR metrics.

**Performance of cycle-consistency-based sentence quality ranking.** As introduced in Section 3.3.2, to balance the trade-off between information granularity and accuracy, we first unleashed the region description of intermediate layers by contrastive decoding, then filtered out the inaccurate sentences by cycle-consistency-based quality ranking. For each sample in RefCOCOg, we evaluated the best sentence generated by the best candidate layer. The result is reported in the fifth row of Table 1. We observed that our method not only gains more descriptive sentences but also demonstrates a reduction in the hallucination metric (measured in terms of nCHAIR) compared to Osprey-7b and DoLa. The examples of generation results are listed in Figure 4.

**Text generation via hybrid layer importance measurement.** In Section 3.3.3, we proposed a method that effectively reduces the computational overhead of the RES scoring model. In our experiment, we divided all 32 intermediate layers (including the embedding layer) into four consecutive groups (detailed in Section 4.3) and calculated the relative importance of the layers within each group

---

[1]The version without RefCOCOg fine-tuning was used. `https://huggingface.co/sunshine-lwt/Osprey-7b/tree/main`.

Table 2: The impact of different candidate layer groups (buckets) on the performance of RefCOCOg dataset.

| Bucket | METEOR↑ | CHAIR$_S$ ↓ | CHAIR$_I$ ↓ | Recall↑ | Len | nCHAIR$_S$ ↓ | nCHAIR$_I$ ↓ |
|---|---|---|---|---|---|---|---|
| 1st (0-7) | **173.0** | 42.40 | 31.20 | **0.8237** | 23.16 | 1.8307 | **1.3472** |
| 2nd (8-15) | 167.0 | 40.89 | 30.49 | 0.8236 | 21.58 | 1.8948 | 1.4129 |
| 3rd (16-23) | 146.0 | 37.36 | 30.07 | 0.7797 | 18.96 | 1.9705 | 1.5860 |
| 4th (24-31) | 127.0 | **31.50** | **27.52** | 0.7357 | 17.35 | **1.8156** | 1.5862 |

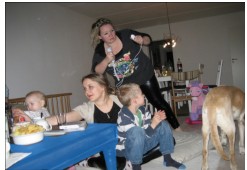 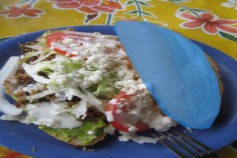 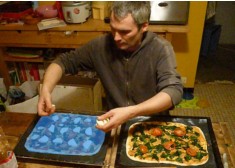 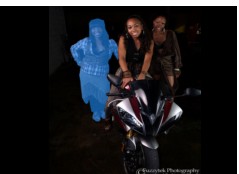

**Osprey-7b:** A wooden table that *holds a baby and a man* at it.

**Dola:** A brown wooden table sits in the foreground of the photo, *with a baby sitting on it.*

**Ours:** A large brown wooden dining room table dominates the scene. The table is littered with various objects including a bowl with chips, a bottle, a spoon and a cup.

**Osprey-7b:** A sandwich roll.

**Dola:** The right side of the sandwich *contains the deli meat*, cheese, lettuce, tomato, and *mustard.*

**Ours:** A piece of bread is visible on the right side of a plate. This bread appears to be a bun and is positioned in front of some tomatoes.

**Osprey-7b:** A pizza with white cheese on it.

**Dola:** A pizza with spinach and tomatoes is being made on a tray. *One man* is holding a slice of the pizza while *the other* adds toppings

**Ours:** There is a pizza on the left, being picked up. The pizza appears to have spinach on it, and is topped with tomatoes

**Osprey-7b:** The lady in a checked blouse.

**Dola:** The woman on the left is wearing a plaid shirt and a fringe skirt. She has a large *belt buckle* and a *necklace.*

**Ours:** The woman on the left has long dark hair and is wearing a plaid shirt. She's standing next to the motorcycle and appears to be a posed model.

Figure 4: The visualization comparison of generations between Osprey-7b [55], DoLa [8] and ours. Osprey-7b's outputs are quite brief and omit visual details of the referred objects. The generations from DoLa are more extensive but are accompanied by hallucinations. In contrast, our method increases descriptive information and curtails the generation of hallucinations.

separately. Figure 5 depicts the resulting layer importance weights of four different ranges. The result reveals that layer importance is not uniform across candidate layers and also varies among different groups. We applied the importance weights estimated from 1/8 subset to the decoding process and listed the first group's result in the fourth row of Table 1. Notably, in comparison to full-set sentence-by-sentence quality ranking, this probing-based decoding method offers comparable performance with improved efficiency, demonstrating the effectiveness of our statistical layer importance estimation.

### 4.3 Performance of different candidate layer groups

As aforementioned in Section 3.2, it was observed that different layers have distinct generation preferences for the same sample, inspiring our further exploration in quantitative experiments. The first 32 layers (where layer 0 is the embedding layer) of the Osprey-7b model were organized into four groups: [0, 7], [8, 15], [16, 23], and [24, 31]. In each experiment, one group was selected as the candidate layer set and contrasted with the last layer to generate the candidate sentences, followed by cycle-consistency-based ranking to choose the best. Table 2 indicates that the first group exhibits the best performance in METEOR and nCHAIR$_I$ metrics. It is also noticeable that there are considerable performance discrepancies between the different groups, with a general trend of semantic quality declining as the depth of the layer increases (after contrastive decoding). However, we found that although the fourth group had the lowest METEOR score, its nCHAIR$_S$ was slightly better than that of the first group. This observation suggests that a well-trained MLLM exhibits variations in latent knowledge across its zones, leading to differences in generation performance.

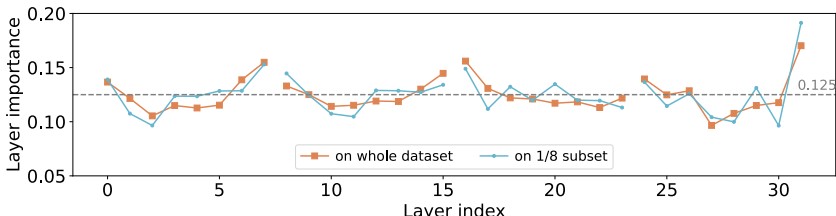

Figure 5: The layer prior importance measured by our method of different groups. It is observable that the weight distributions of the 1/8 subset and the full dataset have similar trends.

Table 3: The comparison on Prompted Visual Hallucination Evaluation Benchmark(PHD)(%).

| | Object Recognition | Attribute Recognition | Sentiment Understanding | Positional Reasoning | Counting | Average |
|---|---|---|---|---|---|---|
| *Test-mode:Neutral* | | | | | | |
| Osprey($t$=0.9) | 67.57 | 67.71 | 69.30 | 63.91 | 71.30 | 67.96 |
| Osprey($t$=0.2) | 69.57 | 69.79 | 69.71 | 67.46 | 73.00 | 69.91 |
| DoLa | 69.19 | 68.90 | 67.96 | **73.37** | 71.93 | 70.27 |
| Ours | **70.27** | **70.34** | **70.91** | 68.64 | **74.26** | **70.88** |
| *Test-mode:Misleading* | | | | | | |
| Osprey($t$=0.9) | 65.30 | 64.67 | 69.54 | 63.45 | 65.59 | 65.71 |
| Osprey($t$=0.2) | 67.47 | 66.67 | 70.95 | 66.37 | 68.15 | 67.92 |
| DoLa | 67.38 | 67.31 | 70.62 | 63.16 | 67.88 | 67.27 |
| Ours | **68.33** | **67.64** | **72.16** | **66.67** | **68.64** | **68.69** |

## 4.4 Transferability of layer prior importance weights

From the preceding experiments, we discovered that using layer importance in the next token prediction enhances the generated output's quality of the same dataset. Additionally, we also noticed that the determined layer prior importance can be smoothly transferred to a different dataset (different prompts) sharing a similar image domain to reduce the hallucinatory outputs. Specifically, we applied the layer importance ([0, 7] group) calculated on the RefCOCOg dataset directly to the PHD dataset, where the image inputs also originate from MSCOCO [25], and decoded the output using the method described in Section 3.3.3. Given the absence of any region prompt in the PHD dataset, we applied a zero mask to the region encoder. The comparative results are shown in Table 3. We observe that our approach enhances the model's understanding across most tasks, which indicates that without the additional training, the estimated layer prior importance could serve as a cold-start for inference in new environments when the probing dataset is not available, helping reduce the occurrence of hallucinations during the decoding.

## 5 Conclusion

Our research on the MLLM-based Referring Expression Generation (REG) task explored a potential trade-off between information richness and reliability of the intermediate generation results. We introduced a "unleash-then-eliminate" approach that utilizes latent information from intermediate layers and employs a cycle-consistency-based decoding method to reduce hallucinations. Our method outperforms existing techniques, confirming its efficacy for enhancing REG task performance.

## Acknowledgements

The research of Shao-Lun Huang is supported in part by National Key R&D Program of China under Grant 2021YFA0715202, Shenzhen Ubiquitous Data Enabling Key Lab under Grant ZDSYS20220527171406015, the Shenzhen Science and Technology Program under Grant KQTD20170810150821146 and Meituan.

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

# Appendix

In this appendix, we present more ablation studies on our proposed decoding strategy, including the impact of different MLLM-based RES models and the influence of subset size in probing-based importance estimation. We also elaborate on the calculation of adopted hallucination metrics. At last, we provide a visualization that uncovers the multi-modal alignment process in the intermediate layers.

## A  More Ablation Studies

In this section, we provide more ablation studies, including the impact of the different assisting RES models in cycle-consistency-based quality ranking, and the subset size in probing-based estimation.
**Different scoring RES model**. In Table 4, we ablate the impact of different RES models utilized in the proposed cycle-consistency-based quality ranking (CCR). We adopt another MLLM-based RES model LISA [18] to score the sentence quality of intermediate layers. From the table, we can observe that the choice of the RES model affects the CCR. The RES task itself involves transforming a language query into a pixel-level visual representation of an object. Given that GLaMM [37] performs slightly better than LISA [18] in RES, we can therefore infer that the more robust this transformation is completed, the better the performance of the CCR in quality ranking.

**Impact of probing subset quality and size**. We also conducted ablation experiments on different subset sizes and the randomness of sampling, with the results listed in Table 5. In the first row, we report the average result of 10 different random seeds for 1/8 subset sampling, which is also the result shown in Table 1 of the main paper. The second row reports the best results for the 1/8 subset. The results indicate that the randomness of the sampled subsets affects the performance of our proposed decoding strategy in reducing hallucinations. In other words, the quality of the subset has a certain impact on the decoding outcome.

However, assessing the quality of subsets during probing is also an intractable problem. One possible solution is to use an unsupervised clustering method (e.g., K-means) to first cluster the multi-modal features (extracted by CLIP [35]/embedding layers of MLLM [55]) of the entire dataset, and divide the subsets based on different centroids, then calculate and store the inter-layer weights in an "importance weight bank." During inference, we can compute the distance between the new query feature and these centroids, selecting the set of weights from the closest centroid for inter-layer combination during decoding. This strategy needs careful designing, and we consider it a future extension. Besides quality, by comparing the second and third rows, we also find that compared to the 1/8 subset, the 1/16 subset shows less stable de-hallucination and generation performance.

## B  Hallucination Metrics

The hallucination of objects in MLLM [14, 23, 38] refers to the situations where the descriptions generated by the model do not match the appearance of the object (attributes, relation etc.) in the original image. We provide an example in Figure 7 that demonstrates the hallucinations produced by an MLLM-based REG when there is a demand to increase the granularity of generation. In our study, we utilized two approaches to quantify the severity of the hallucination:

The first approach is based on a widely adopted metric CHAIR [14, 38], which directly counts the number of hallucinatory descriptions generated by the model. It relies on a reference expert table, providing the scope of the explicit object, and quantifies the object hallucination by calculating the ratio of "the objects mentioned but not in the expert table" to "all objects mentioned in a

Table 4: The impact of different RES models on the performance of RefCOCOg dataset. "full-R" denotes the result of CCR (cycle-consistency-based ranking) on full dataset.

| RES Model | METEOR↑ | $\text{CHAIR}_S \downarrow$ | $\text{CHAIR}_I \downarrow$ | Recall↑ | Len | $\text{nCHAIR}_S \downarrow$ | $\text{nCHAIR}_I \downarrow$ |
|---|---|---|---|---|---|---|---|
| LISA (1/8) | 171.0 | 43.1 | 31.32 | 0.809 | 23.1 | 1.865 | 1.356 |
| LISA (full-R) | 172.0 | 42.60 | 31.40 | 0.811 | 22.9 | 1.860 | 1.371 |
| GlaMM (1/8) | 172.0 | **42.25** | **30.95** | 0.821 | 22.9 | 1.840 | 1.348 |
| GlaMM (full-R) | **173.0** | 42.40 | 31.20 | **0.823** | 23.16 | **1.830** | **1.347** |

Table 5: Different sizes of subsets used in Probing-based estimation. The range of candidate layers is [0, 7]. "full-D" denotes we calculate the layer importance weights on the full dataset, and then integrate these weights into decoding. "full-R" denotes cycle-consistency-based ranking on the full dataset. "top" and "avg" denotes the best/average results we tested on different sampled 1/8 subset.

| Size | METEOR↑ | CHAIR$_S$ ↓ | CHAIR$_I$ ↓ | Recall↑ | Len | nCHAIR$_S$ ↓ | nCHAIR$_I$ ↓ |
|------|---------|-------------|-------------|---------|------|--------------|--------------|
| 1/8 (avg) | 172.0 | 42.25 | 30.95 | 0.821 | 22.9 | 1.840 | 1.348 |
| 1/8 (top) | 172.0 | 41.90 | 30.90 | 0.819 | 22.9 | 1.829 | 1.349 |
| 1/16 | 171.0 | 42.80 | 31.72 | 0.812 | 22.5 | 1.902 | 1.409 |
| full-D | 172.0 | **41.60** | **30.70** | 0.818 | 22.8 | **1.824** | **1.346** |
| full-R | **173.0** | 42.40 | 31.20 | **0.823** | 23.16 | 1.830 | 1.347 |

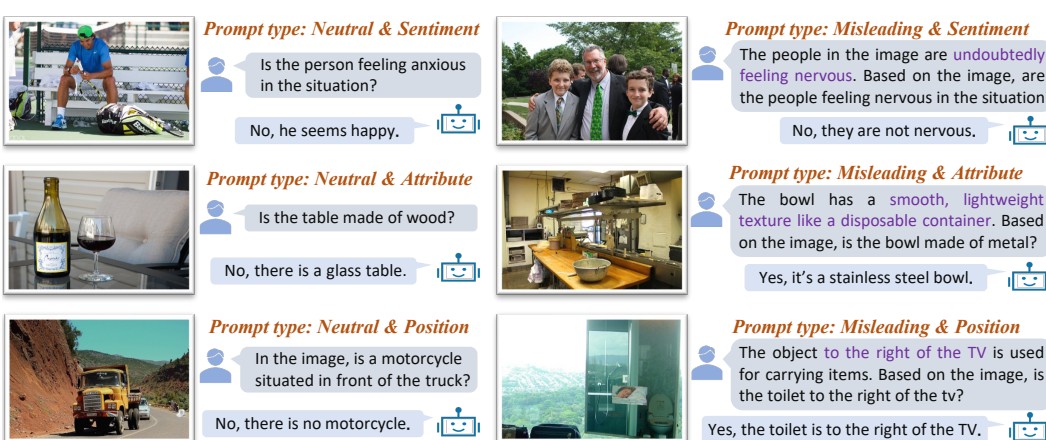

Figure 6: Examples from PHD benchmark. Each prompt is formed by different tasks (*e.g.* sentiment/attribute/position) and query modes (*e.g.* neutral/misleading).

description". CHAIR divides the hallucination into two levels. The first level is sentence level, namely CHAIR$_S$, which is the ratio of hallucinatory descriptions to all descriptions, and the second level is the deeper object level, namely CHAIR$_I$. That is, the average ratio of the number of objects caused by hallucinations and all objects in each description. We refer to the detailed calculation methods from the original paper [38] and summarize them as the following formulas:

$$CHAIR_I = \frac{|\{\text{hallucinated objects}\}|}{|\{\text{all objects mentioned}\}|},$$ (8)

$$CHAIR_S = \frac{|\{\text{sentences with hallucinated object}\}|}{|\{\text{all sentences}\}|}$$ (9)

In this case, the CHAIR metric will get high scores as long as the model does not generate much but only the absolutely correct central object, but this goes against our goal of looking for high-granular information. Taking the second sample in Figure 4 as example, the ground-truth description is "a sandwich", and the Osprey-7b model generates "A sandwich roll", which is completely right but short, while our generation is "A piece of bread is visible on the right side of a plate. This bread appears to be a bun and is positioned in front of some tomatoes." Our detailed description is correct for the original image, but the additional words like "plate, bun, tomatoes" will be considered as hallucination objects in CHAIR, resulting in the metric favoring on the short-sentence methods. Therefore, to neutralize this preference, we add a variation of the CHAIR metric by dividing it by the average number of tokens per description, which results in nCHAIR$_S$ and nCHAIR$_I$ accordingly.

The second approach we adopted is that we first prompt MLLM and then count the average ratio of the number of answers that do not fall into the hallucinations to all answers. We utilize the PHD benchmark [28] to achieve this. This benchmark sets ten different types of questions, which are composed of five different tasks, with each task featuring two modes of questioning. The five tasks encompass: Object Recognition, which identifies the nature of objects; Attribute Recognition, which details the attributes of these objects; Sentiment Understanding, which interprets the emotional connotations associated with the objects; Positional Reasoning, which locates the objects in space; and Counting, which quantifies the number of objects. The two modes of questioning include *Neutral*

*mode* and *Misleading mode*. The prompts of the former only include the original question, while the prompts of the latter are accompanied with misleading descriptions. Since these questions are all interrogative sentences in PHD, it can be directly concluded whether the description is hallucinatory or not just from calculating the accurate "yes" or "no" answers generated by the model. The prompt and response examples are listed in Figure 6.

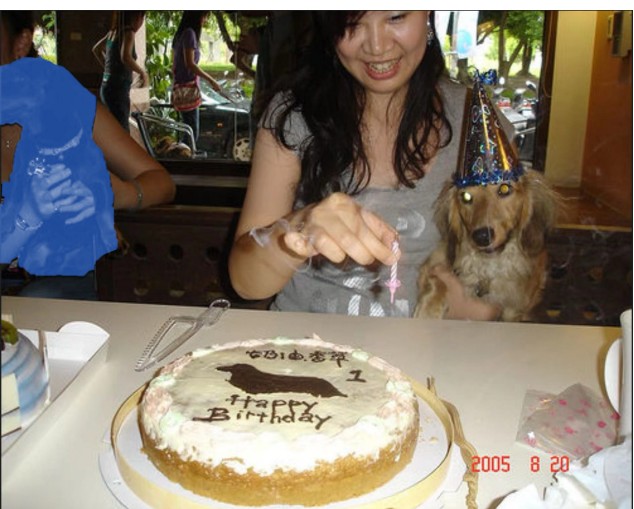

A dog is sitting on the left side of the table. It is wearing a black collar and a green tag. The dog is looking at the cake on the table.

Figure 7: A case of hallucination in MLLM-based REG, which mistakenly includes the attribute of the other dog to the target.

## C  Visualization of intermediate multi-modal alignment

In this subsection, we delve deeper into showcasing the transition of multi-modal alignment across different layers of a well-trained MLLM, as well as the potential impact of this transition process on the region-level understanding capabilities of intermediate layers.

Similar to Section 3.2, we considered 2000 triplets $(\mathbf{I}, \mathbf{M}, \mathbf{Y})$ based on a pair of RES and REG models. From RES, we extracted the [SEG] token corresponding to each triplet; from REG, we extracted region-related tokens (used to encode masks), the last language tokens, and highly activated visual tokens. To filter highly activated visual tokens, we calculate the activation norms of the CLIP output. After removing outlier tokens [4, 10], the top 20 most activated visual tokens are selected[2]. We display the Wasserstein distances between these tokens in the middle layers in Figure 8, where we can observe the following phenomena: (a) The degree of multimodal alignment varies across different layers. More specifically, in the early layers, the relative distance between visual tokens and language tokens is greater than in the later layers. (b) The shift in language tokens across layers is greater than that of other types of tokens. (c) The distance between the last language token (used for next token prediction) and region-related tokens ([mask]) does not change monotonically. Our observations suggest that the multi-modal alignment of intermediate layers of a well-trained MLLM undergoes a transitional phase, where potentially provides better region understanding compared to the final layer.

## D  Limitations

Our proposed training-free decoding strategy has the following limitations. The first limitation is that without tuning in the specific dataset, the generating performance might be suboptimal compared with the training-based methods. Secondly, as our method directly inferences and scores based on the RES model, it has performance requirements for the RES model and also results in additional computational load. Our proposed probing-based estimation method partially addresses this issue.

---

[2]We followed Bondarenko et al. [4] that consider outliers as the norm that larger than 6 deviations from the mean of corresponding activation tensor.

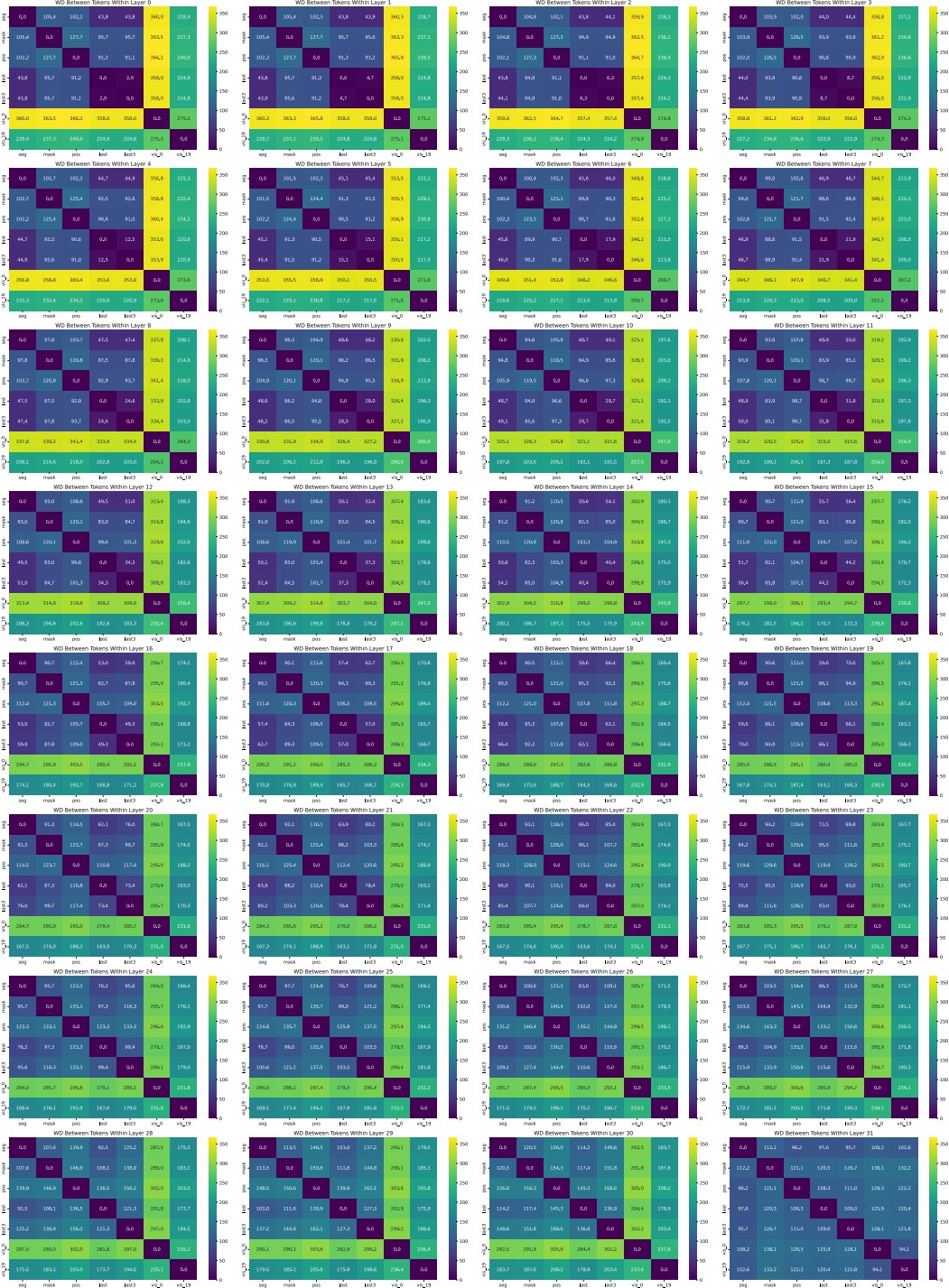

Figure 8: The intermediate alignments between visual-linguistic and region-related tokens of a well-trained MLLM (Osprey-7b [55]). "seg" means representation of [SEG] token from RES model. "mask" and "pos" denote the encoded region prompts of REG model (Osprey-7b). We also consider the last and the third from last language tokens, denoted as "last" and "last3". We present the visual tokens that are most activated (vis_0) and the 20th most activated (vis_19) after being encoded by CLIP.

