# OpenReview forum: "Unleashing Region Understanding in Intermediate Layers for MLLM-based Referring Expression Generation"
_NeurIPS.cc/2024/Conference — NeurIPS 2024 poster_

### Official Review · Reviewer_ikvW · 2024-07-06

**Soundness:** 3
**Presentation:** 3
**Contribution:** 3
**Rating:** 6
**Confidence:** 4

**Summary:**

This paper explores the Multi-modal Large Language Model (MLLM) based Referring Expression Generation (REG) task, which aims to generate unambiguous text descriptions for specific objects or regions in images. MLLM-based REG models tend to suffer from hallucination issues, and there is a trade-off between detailed descriptions and accurate targeting. To address this, a training-free method called "unleash-then-eliminate" is proposed, which elicits latent information in intermediate layers and uses cycle-consistency decoding to reduce hallucinations. Extensive experiments on the RefCOCOg and PHD benchmarks show that this method outperforms existing approaches in both semantic and hallucination-related metrics.

**Strengths:**

1. The observation that the intermediate layers of the current region-level MLLMs sometimes hold more descriptive regional information than the final layer is interesting.
2. The writing throughout the paper is clear and easy to follow. The authors have done a good job in presenting their ideas and methodologies in a manner that is both logical and comprehensible.
3. The experimental results are compelling and demonstrate that the proposed method outperforms existing methods on the newly introduced evaluation metric.

**Weaknesses:**

1. Since the proposed method requires multiple layers for inference to be described, both REG and RES models are required, resulting in a very low efficiency of the method and requiring a lot of additional calculations and memory.
2.  The ablation study is not sufficient. For example, subset used to select the optimal layer in Layer Importance Measurement. What is the Impact of subset size and quality on performance? What is the performance of using  cycle-consistency-based candidate ranking process for whole dataset, i.e., not using Layer Importance Measurement.
3.  What is effect of using different RES models for the proposed method.

**Questions:**

1. What is the effect of \alpha defined in Line 172
2. \mathcal{V} is not defined in Line 171.
3.  The candidate sentence set A use n to represent  the size of candidate layers, while use N-1 in Line 169. So are n and N-1 the same, or do they represent different things?

**Limitations:**

The authors adequately addressed the limitations.

---

> ### Author Rebuttal · Authors · 2024-08-07
>
> Thank you for your attention to our work and your positive acknowledgment of our idea. We have open-sourced the code. We will address your concerns below.
>
> **1. low efficiency.** We would like to hightlight more about probing-based estimation that is designed to allivate this issue. While the cycle-consistency-based quality ranking incorporates the RES model, adding to the computational load, our probing-based estimation method simplifies this process. By transforming the REG-RES cycle into a straightforward set of importance weights (please refer to Figure A in our common response), we eliminate the necessity for RES in decoding. The proposed method and its resulting importance prior weights can effectively reduces MLLM hallucinations by using combinations of intermediate layers. We have validated this approach in our experiments, as shown in Table 3 of the PHD benchmark. We hope this could partially address your concern.
>
> **2. Ablation study.** Thanks for your precise and valuable feedback. We have reported some subset-related ablation experiments in the table below. The range of candidate layers is [0, 7]. As for the “full-D”, we calculate the layer importance weights on the full dataset, and then integrate these weights into decoding. “full-R” denotes cycle-consistency-based ranking on the full dataset. “top” and “avg” denotes the best/average results we tested on different sampled 1/8 subset.
>
> | Size   | METEOR | CHAIR_S | CHAIR_I | Recall | Len  | nCHAIR_S | nCHAIR_I |
> |--------|--------|---------|---------|--------|------|----------|----------|
> | 1/8 (avg) | 172.0  | 42.25   | 30.95   | 0.821  | 22.9 | 1.840    | 1.348    |
> | 1/8 (top) | 172.0  | 41.90   | 30.90   | 0.819  | 22.9 | 1.829    | 1.349    |
> | 1/16     | 171.0  | 42.80   | 31.72   | 0.812  | 22.5 | 1.902    | 1.409    |
> | full-D   | 172.0  | **41.60**   | **30.70**   | 0.818  | 22.8 | **1.824**    | **1.346**    |
> | full-R   | **173.0**  | 42.40   | 31.20   | **0.823**  | 23.16| 1.830    | 1.347    |
>
> In the first row, we report the average result of 10 different random seeds for 1/8 subset sampling, which is also the result shown in Table 1 of the main paper. The second row reports the best results for the 1/8 subset. The results indicate that the randomness of the sampled subsets affects the performance of our proposed decoding strategy in reducing hallucinations. In other words, the quality of the subset has a certain impact on the decoding outcome.
>
> However, assessing the quality of subsets during probing is also an intractable problem. One possible solution is to use an unsupervised clustering method (e.g. K-means) to first cluster the multimodal features (extracted by CLIP or embedding layers of MLLM) of the entire dataset, and divide the subsets based on different centroids, then calculate and store the inter-layer weights in an "importance weight bank." During inference, we can compute the distance between the new query and these centroids, selecting the set of weights from the closest centroid for inter-layer combination during decoding. This strategy needs careful designing, and we consider it a future extension. Besides quality, by comparing the second and third rows, we also find that compared to the 1/8 subset, the 1/16 subset shows less stable de-hallucination and generation performance.
>
> **3. Different RES model.**
> We appreciate for the detailed comments. In the following Table , we ablate the impact of different RES models utilized in the proposed cycle-consistency-based quality ranking (CCR). We adopt another MLLM-based RES model LISA to score the sentence quality of intermediate layers.
>
> | RES Model   | METEOR | CHAIR_S | CHAIR_I | Recall | Len   | nCHAIR_S | nCHAIR_I |
> |-------------|--------|---------|---------|--------|-------|----------|----------|
> | LISA (1/8)  | 171.0  | 43.1    | 31.32   | 0.809  | 23.1  | 1.865    | 1.356    |
> | LISA (full-R) | 172.0 | 42.60   | 31.40   | 0.811  | 22.9  | 1.860    | 1.371    |
> | GlaMM (1/8) | 172.0  | **42.25**   | **30.95**   | 0.821  | 22.9  | 1.840    | 1.348    |
> | GlaMM (full-R) | **173.0**| 42.40   | 31.20   | **0.823**  | 23.16 | **1.830**    | **1.347**    |
>
> From the table, we can observe that the choice of the RES model affects the CCR. The RES task itself involves transforming a language query into a pixel-level visual representation of an object. Given that GLaMM performs slightly better than LISA in RES, we can therefore infer that the more robust this transformation is completed, the better the performance of the CCR in quality ranking.
>
> **4. Explanation of symbols.**
>  Thanks for the question and we apologize for the confusion. $\alpha$ is a hyperparameter in [0, 1] that truncates the next token distribution of $p(y_t|y_{<t})$. It helps split out the tokens whose score is lower than a proportion of the highest score. Larger $\alpha$ entails more aggressive truncation, keeping only high-probability tokens, whereas smaller $\alpha$ allows tokens of lower probabilities to be generated [Li et al., arXiv:2210.15097].
>
> $\mathcal{V}$ in Line 171 denotes the vocabulary set.
>
> $n$ and $N-1$ are different. $N$ denotes the number of layers of MLLM model, 32 in our case, and $N-1$ represents the number of remaining layers except the final layer. The lowercase $n$ is dynamic and refers to the number of candidate layers in experiments, which we chose as 8 in our implementation.

---

> ### Author Response · Authors · 2024-08-12
>
> Thank you once again for your insightful comments. We deeply appreciate your guidance and found great resonance with your perspectives. With the discussion period nearing completion in less than two days, please feel free to share any final comments at your convenience.

---

> > ### Comment · Reviewer_ikvW · 2024-08-13
> >
> > The author did address some of my concerns, and I keep my initial rating.

---

### Official Review · Reviewer_85oQ · 2024-07-07

**Soundness:** 2
**Presentation:** 1
**Contribution:** 2
**Rating:** 5
**Confidence:** 3

**Summary:**

This paper presents an approach for improving the accuracy and richness of referring expression generation (REG) by leveraging the descriptive potential of intermediate layers in Multi-modal Large Language Models (MLLMs). The method employs a cycle-consistency-based decoding strategy to reduce hallucinations and improve descriptive quality. The proposed method is evaluated on the RefCOCOg and PHD benchmarks, demonstrating superior performance over existing methods.

**Strengths:**

The "unleash-then-eliminate" strategy and the use of intermediate layers for generating more detailed descriptions is innovative to me.

**Weaknesses:**

The proposed method introduces additional complexity, particularly in the decoding process. While effective, the cycle-consistency-based approach may increase computational overhead, which could limit its applicability in real-time or resource-constrained environments.

The related work section lacks depth in its analysis and could benefit from a more thorough review of recent advancements, particularly in hallucination mitigation techniques.

Some terms and notations, such as Q, H and W in Section 3.1 , are not clearly defined in the context of the paper, which can lead to confusion. Providing explicit definitions and clarifications would improve readability.

**Questions:**

You use PCA for dimensionality reduction and Wasserstein distances for comparing features. Can you elaborate on why these specific techniques were chosen and how they contribute to the effectiveness of your method compared to other possible choices?

Your method aims to reduce hallucinations in generated descriptions. Can you provide more details on how you quantify hallucinations and the specific improvements observed with your method compared to baselines? Are there any cases where your method still struggles with hallucinations?

Is there any Transformer-based region encoder?

**Limitations:**

The authors have not adequately addressed the limitations and potential negative societal impacts of their work.

---

> ### Author Rebuttal · Authors · 2024-08-07
>
> Thanks for your feedback. We have made the code open source at the link attached to the abstract. We will address your concerns below.
>
> **1. Increase computational overhead.** Thanks for the comments.
> In the cycle-consistency-based quality ranking, the RES model is incorporated as an auxiliary, leading to increased computational overhead. However, we need to emphasize that our proposed probing-based estimation method simplifies the cycle between REG and RES into a set of importance weights (refer to Figure A in common response). It removes the need for RES during decoding, and mitigates the MLLM hallucinations through combinations of intermediate layers. This result has been demonstrated in our experiments, on the Table 3 PHD benchmark.
>
> **2. More related work.**
> Thank you for your suggestion. Based on your advice, we would like to summarize some related works on inference-time decoding strategies for mitigating hallucination in large language models (LLMs). Specifically, our work represents an inference-time decoding strategy to mitigate hallucinations, which utilizes the latent knowledge within the intermediate layers without additional training. There are more details in the code link. We hope this could help you better understand the positioning of our work.
>
> **3. Undefined terms.** Thanks for your feedback. HW denotes the flattened encoded visual token numbers. Q denotes the number of tokens that the model used to encode the region prompt.
>
> **4. The choices used in visualization.**
> Thanks for the question. We first observed through the early-exit method that outputs from different intermediate layers tend to vary, and some may even contain more descriptive content than the final layer. Subsequently, we attempted to quantify how much potential different intermediate layers have for better region understanding. PCA and Wasserstein distances are two common analytical methods used. PCA reduces the dimensionality of high-dimensional latent features while preserving the principal components, which not only speeds up the visualization process but also allows features from different dimensions to be reduced to the same dimension for distance calculation. We use Wasserstein distances as a measure of distance because it evaluates the distance between two distributions from the perspective of transportation cost. Compared to other methods used to measure distances between distributions, it requires fewer samples [M. Arjovsky et al., ICML2017].
>
> **5. How to quantify hallucinations.**
> Thanks for the question. According to your suggestion, we would like to provide a detailed elucidation of the hallucination metric calculation and summarize it as follows:
> In our study, we utilized two approaches to quantify the severity of the hallucination:
> 1. The first approach is based on a widely adopted metric CHAIR, which directly counts the number of hallucinatory descriptions generated by the model. We use it for the sentences of REG task. It relies on a reference expert table, providing the scope of the explicit object, and quantifies the object hallucination by calculating the ratio of "the objects mentioned but not in the expert table" to "all objects mentioned in a description". The result is reported in Table 1 & 2.
> 2. The second approach we adopted is prompting MLLM and counting the average ratio of the number of answers that do not fall into the hallucinations to all answers. We utilize the PHD benchmark (a challenging extension of POPE) to achieve this. Similar to POPE, since prompted questions are all interrogative sentences in this benchmark, it can be directly concluded whether the description is hallucinatory or not just from calculating the accurate "yes" or "no" answers generated by the model. The result is reported in Table 3. For more details, please refer to the code link.
>
> **6. The specific improvements and failure case.**
> 1. **specific improvements:**
> By extracting information from the intermediate layers of MLLMs, our method generates sentences that are more descriptive while reducing hallucinations. Specifically, Table 1 shows an increase in the METEOR metric, indicating higher quality sentence generation, and an improvement in the CHAIR series metrics, showing a reduction in hallucinations. Table 3's improvements on the PHD benchmark further demonstrate that our decoding strategy effectively mitigates hallucination.
>
> 2. **failure case:**
> Thank you for the question, and we provide a failure case as Figure C-(6) in the common response. In this case, the referred object is the black dog on the left. It can be seen that during the generation process, the model initially describes the referred object in detail, but eventually a hallucination occurs, mistaking the state of another dog (looking at the cake) for the state of the referred dog. This also illustrates the unique challenges of region-level understanding.
>
> **7. Is there any Transformer-based region encoder?** Thank you for the question. In the context of our paper, the region encoder is used to encode region prompts for the MLLM. To our knowledge, MLLMs that perform REG have not yet employed transformer blocks for encoding region prompts. Shikra [K. Chen et al., arXiv:2306.15195] converts regions into coordinates, represented using natural language numbers. Ferret [H. You et al., ICLR2024] uses CNN blocks and point sampling for encoding. GPT4ROI [S. Zhang et al., arXiv:2307.03601] and GLAMM [H. Rasheed et al., CVPR2024] employ ROI pooling methods, and Osprey [Y. Yuan et al., CVPR2024] utilize CNN blocks for multi-scale mask encoding. We speculate that such designs may aim to reduce computational complexity, as transformer modules require a larger amount of data to fit. Given the vast knowledge underlying pre-trained visual encoders (CLIP) and the MLLM base (LLAVA), the region prompts need to align with the existing knowledge. A lightweight region encoder could reduce the optimization difficulty.

---

> ### Author Response · Authors · 2024-08-12
>
> It is great to know that our responses are helpful, and thank you for your positive feedback and support for this work! Your suggestions have helped improve this work and we will incorporate them in the revised manuscript.

---

### Official Review · Reviewer_MMWB · 2024-07-12

**Soundness:** 2
**Presentation:** 1
**Contribution:** 1
**Rating:** 4
**Confidence:** 4

**Summary:**

The paper aims to strike a balance between detailed description and precise captioning when using multimodal large language models (MLLM) in the task of referring expression generation (REG). A key observation is that the output of a Referring Expression Segmentation (RES) model should be consistent with the input of a REG model.

Based on this observation, a training-free method has been proposed, called 'unleash-then-eliminate,'  which adopts an 'elicit-then-eliminate decoding' strategy. Captions are generated using contrastive decoding (Li et al., arXiv:2210.15097) and then fed into a RES model to produce corresponding masks. The Intersection over Union (IoU) between the masks tagged by the RES model and the input of the REG model is leveraged to select the candidate layer along with the generated caption. Additionally, a Probing-based Importance Estimation method is proposed to accelerate the decoding process.

The generation quality of the model is evaluated using the METEOR score on the RefCOCOg dataset. Additionally, the model's performance in avoiding hallucinations is assessed with the CHAIR and PHD metrics. The model proposed in this paper outperforms both Osprey and Dora not only in terms of generation quality but also in terms of adequacy.

**Strengths:**

1. The problem definition and motivation are clear. The paper indicates that a balance between detailed description and accurate targeting is necessary when using MLLM.

2. The model architecture is clear. The proposed model utilizes contrastive decoding to unleash the information in intermediate layers for generating captions. It indirectly evaluates the quality of these layers by assessing the masks generated by a RES model in relation to the captions. Additionally, a Probing-based Importance Estimation method is proposed to expedite the decoding process.

**Weaknesses:**

1. The writing is disorganized and lacks clarity. For example, in the first sentence "Referring expression generation (REG) Yu et al. [2016], Mao et al. [2016], Hu et al. [2016], Yu et al. [2017], Luo et al. [2020], Ding et al. [2021], Tanaka et al. [2019] is a task to", the reference is mixed with words, making it difficult to follow.

2. The proposed approach is technically simple. The architecture seems simply a combination of REG and RES models.

3. The motivation is to achieve a trade-off between the granularity and accuracy of captioning by selecting different intermediate layers. However, the experiments only show that the proposed model outperforms other models, as indicated in Table 1. Furthermore, Table 2 reveals that the first bucket of layers achieves the best scores in both METEOR, which relates to generation quality, and nCHAIR_I, which relates to the avoidance of hallucinations. This finding contrasts with the initial proposal.

4. The answer to `Open access to data and code` is [Yes], but there's only a `placeholder` found in the anonymous repo,  noted as footprint in the first page of paper.

**Questions:**

1. What is the final architecture of the proposed model with `Probing-based Importance Estimation` ?

2. What are `the number of parameters` of the proposed models (with and without Probing-based Importance Estimation)?

3. What are the metrics corresponding to those types (i.e. Object Recognition...) in table 3? What is the difference between `neutral-question mode` and `misleading mode`?

**Limitations:**

Yes.
The authors claim that since the model has not been tuned on a specific dataset, the generating performance is suboptimal compared to training-based methods.

---

> ### Author Rebuttal · Authors · 2024-08-07
>
> Thank you for your detailed review and constructive comments. The code is accessible through the link indicated in the abstract. We noticed that there might have been some misunderstandings regarding to our proposed "Probing-based importance estimation" method, and we hope this response will better convey our decoding strategy to you.
> We will address your concerns below.
>
> **1. Reference format.** Thanks for the suggestion. We have changed it to numerical style in the revision. Additionally, we would like to summarize some related works on inference-time decoding strategies for mitigating hallucination in large language models (LLMs), hoping this can better help you position our work. Specifically, our work is in line with CCS [C. Burns et al., ICLR2022], ITI [K. Li et al., NeurIPS2023], and DoLa [Y.-S. Chuang et al., ICLR2024], which are all aiming to utilize the existing knowledge in trained models through representational operations to enhance the authenticity of the output. Compared to these works, we uncover the intermediate layers of multimodal LLMs and, as far as we know, are the first to propose using a combination of intermediate layers for region-level caption generation. There are more details in the code link.
>
> **2. The architecture seems simply a combination of REG and RES models.** Structurally, the cycle-consistency-based ranking method we proposed indeed relies on the combination of REG and RES. RES serves as an auxiliary tool we utilize to filter the intermediate layer information unleashed afterward.
> Additionally, we would like to highlight some key points beyond the architecture:
> 1. We designed an inference time decoding strategy, which can unleash the latent knowledge inside the intermediate layers without additional training.
> 2. The key observation is: compared with the output of the last layer, the sentences of intermediate layers can sometimes provide more discriminative descriptions for the referred object, which motivates us to design a cycle between REG and RES to localize the desired information.
> 3. Besides the REG and RES cycle process, we also proposed a probing-based estimation method that can obviate the need for RES, directly blending estimated weights into the existing model to enhance its anti-hallucination efficacy.
>
> **3. The trade-off between granularity and accuracy.** Thank you for your detailed comment and we apologize for any confusion caused. We would like to clarify that the trade-off discussed is inherent to the REG task itself. The more extensively a model describes, the higher the likelihood of errors, as demonstrated by the CHAIR-related metrics in Table 1. As descriptions become lengthier, both $\text{CHAIR}_I$ and $\text{CHAIR}_S$ metrics deteriorate, indicating an increase in hallucinations during detailed description generation. To alleviate this issue, we unleash information from the intermediate latent knowledge to broaden the space of choice. Then a RES model as a listener is used to evaluate and select proper output. This "unleash-then-eliminate" approach helps us get rid of the intrinsic constraint of detailedness and accuracy. Table 1 demonstrates the efficacy of our proposed method. In Table 2, our goal is to investigate the effects of candidate layers from different zones using our method, and to illustrate the properties of various regions within a trained MLLM. We have not explored such a trade-off across different layers.
>
> **4. Open source.** Thank you for your interest in our implementation. The code has been uploaded.
>
> **5. Final architecture of Probing-based estimation, and the number of parameters.** We have noticed that there might be some misunderstandings about the approach we have proposed. We have introduced **a decoding strategy for inference-time**, not a new model that requires training, hence **no new parameters** have been added.
> The Probing-based estimation method involves the following steps:
> 1. First, sample a subset as the probing set.
> 2. On this probing set, perform cycle-consistency-based ranking and calculate scores for each sample in each layer.
> 3. Calculate an inter-layer importance weight on this subset, which is mainly based on frequency counting of each candidate layer being the optimal layer.
> 4. Combine the importance weight prior with the J-S divergence to represent the probability of each layer being selected (the higher the importance, the more likely it is to be chosen).
> 5. Sample from this hybrid importance distribution to decide which layer to use for decoding the next token, until the generation finishes.
> This approach compresses the “cycling” process into a set of importance weights, thus facilitating convenient decoding. We would like to emphasize that the experiments in Table 3 have demonstrated that such a set of weights can be directly applied to MLLM to alleviate hallucinations, without the need for RES models and the cycling process, enhancing the practicality of our method. We also provide a visualization of importance weights in **Figure A** (listed in common response).
>
> **6. The metrics of Table 3.** These refer to the prompt types of PHD benchmark. For better illustration, we provide some prompt and generation examples in **Figure B** (in common response). This benchmark sets ten different types of questions, which are composed of five different tasks, with each task featuring two modes of questioning. We refer you to the 'Datasets and Metrics' section and the newly added Appendix C for details on the different tasks. The two modes of questioning include Neutral mode and Misleading mode. The prompts of the former only include the original question, while the prompts of the latter would accompanied by misleading descriptions. More details are available in the code link.

---

> > ### Comment · Reviewer_MMWB · 2024-08-13
> >
> > The author did address some of my concerns and I'll raise my score from 3 (Reject) to 4 (Borderline reject).

---

> > > ### Author Response · Authors · 2024-08-14
> > >
> > > We really appreciate your acknowledgment of our responses to the initial comments, and the adjustment of the rating. We would be grateful for the opportunity to discuss any remaining concerns that could be addressed in our manuscript further. Thank you very much for your time!

---

> ### Author Response · Authors · 2024-08-12
>
> Thank you once again for your valuable comments and thorough review. We have tried our best to clarify the concerns on the paper. Your detailed feedback has enhanced the clarity of our explanations. With the discussion period nearing completion in less than two days, we would appreciate it if you could let us know if any aspects remain unclear. We truly appreciate this opportunity to improve our work and shall be most grateful for any feedback you could give us.

---

### Official Review · Reviewer_adaN · 2024-07-12

**Soundness:** 3
**Presentation:** 3
**Contribution:** 3
**Rating:** 8
**Confidence:** 4

**Summary:**

The paper addresses the Referring Expression Generation (REG) task using Multi-modal Large Language Models (MLLMs), identifying the key challenge as the trade-off between generating detailed descriptions and accurately targeting the referring objects, which often leads to hallucinations—the inclusion of incorrect or spurious information in the generated text. To address this issue, the authors propose a training-free "unleash-then-eliminate" method that leverages the intermediate layers of MLLMs and employs a cycle-consistency-based decoding strategy to mitigate hallucinations. The proposed approach is validated through extensive experiments on the RefCOCOg and PHD benchmarks, demonstrating superior performance compared to existing techniques.

**Strengths:**

1. The paper is in well-written, which makes it easy to understand.

2. The proposed "unleash - then - eliminate" method is training-free, which can avoid the need for additional data and training, reducing the complexity and cost of the model.

3. I like the idea of utilizing the latent information in the intermediate layers of the current region-level MLLMs, which is often overlooked but contains more descriptive regional information.

4. The cycle-consistency-based decoding method helps to alleviate the production of hallucinations in the generated sentences, improving the accuracy and reliability of the model's output. The hybrid layer importance measurement strategy not only increases the decoding speed but also maintains the ability to mitigate hallucinations, achieving a good balance between efficiency and performance.

5. The method shows superior performance compared to existing methods on both semantic and hallucination - related metrics in the experiments, demonstrating its effectiveness.

**Weaknesses:**

1. Without tuning in the specific dataset, the generating performance of the method might be suboptimal compared to the training - based methods.

2. The methods used in the paper, such as cycle ranking, may introduce additional computational load, which could affect the per-sample decoding speed. Although the proposed strategy helps to alleviate this issue, it may not completely solve it.

3. The method assumes that the RES model can accurately estimate the region-aware descriptive performance of the captions generated by the candidate layers. However, the RES model may also have its own limitations and errors, which could affect the final evaluation results.

**Questions:**

N/A

**Limitations:**

refer to weaknesses

---

> ### Author Rebuttal · Authors · 2024-08-07
>
> Thank you for your positive acknowledgment of our work. We are glad to notice your interest in the latent information of the intermediate layers. We hope our responses below will partially address your concerns.
>
> **1. Suboptimal compared to the training-based methods.**
>
> Yes, we proposed an inference-time decoding strategy that utilizes existing knowledge in trained models through representational operations to enhance the authenticity of the output. If the training is allowed, there are two potential areas within our framework where performance could be improved through training:
> 1. If more descriptive datasets are available, the selection among intermediate layers could depend not only on the RES model but also on the ground truth sentence.
> 2. A linear layer could be designed to learn the mapping from different scores across layers to the importance weights of layers.
> We still need to emphasize that although training-based methods might yield better results, as you mentioned, the proposed "unleash-then-eliminate" method is training-free, which also avoids the need for additional data and training.
>
> **2. Computation load.**
>
> Thanks for the feedback. We agree that the computational intensity of the cycle ranking could potentially affect the decoding speed. This is an inherent challenge when integrating more complex algorithms to improve the accuracy of the generation. We would like to highlight that in probing-based estimation, we compress the cycle process into a set of layer importance weights which could directly be merged into the original decoding process, thus enhancing the practicality of our method.
>
> **3. The RES model is imperfect.**
>
> Thanks for the precise comments. We acknowledge that our method depends on the performance of the RES model. We hope that more powerful RES models in the future will help alleviate this limitation. In the table below, we have ablated the impact of the existing MLLM-based RES on our method. Given that GLaMM performs slightly better than LISA in RES, we can therefore infer that the more robust the RES model is, the better the performance of cycle-consistency-based in quality ranking. “full-R” denotes cycle-consistency-based ranking on the full dataset.
>
> | RES Model   | METEOR | CHAIR_S | CHAIR_I | Recall | Len   | nCHAIR_S | nCHAIR_I |
> |-------------|--------|---------|---------|--------|-------|----------|----------|
> | LISA (1/8)  | 171.0  | 43.1    | 31.32   | 0.809  | 23.1  | 1.865    | 1.356    |
> | LISA (full-R) | 172.0 | 42.60   | 31.40   | 0.811  | 22.9  | 1.860    | 1.371    |
> | GlaMM (1/8) | 172.0  | **42.25**   | **30.95**   | 0.821  | 22.9  | 1.840    | 1.348    |
> | GlaMM (full-R) | **173.0**| 42.40   | 31.20   | **0.823**  | 23.16 | **1.830**    | **1.347**    |

---

> > ### Comment · Reviewer_adaN · 2024-08-13
> >
> > Thanks for the response. Authors made a good rebuttal. After carefully reading it, I find all my concerns have been properly solved. I also would like to say,  every research paper may have some imperfections, but I am more concerned with whether the paper offers new insights that can drive further exploration in the field. The presence of minor flaws does not diminish the value of a work that makes a meaningful contribution. From my perspective, the quality of this work has satisfactorily met the established requirements for acceptance. Thus, I would like to give STRONG ACCEPT as my final score.

---

> ### Author Response · Authors · 2024-08-12
>
> Thank you once again for your insightful comments! We deeply appreciate your precise feedback and truly resonate with your perspectives, which have been instrumental in enhancing our work. With the discussion period nearing completion in less than two days, please feel free to share any final remarks at your convenience.

---

> ### Author Response · Authors · 2024-08-14
>
> Thank you so much for your encouraging feedback and for upgrading your score to a "strong accept." We are truly grateful for your recognition of the efforts made in our rebuttal. It's great to hear that our responses have successfully addressed your concerns. We're committed to continuously improving our work, taking into account your insights and those from other reviewers. Thanks again for your supportive words!

---

### Author Rebuttal · Authors · 2024-08-07

We appreciate the time and effort of all reviewers in reviewing our manuscript. Your insightful feedback has been essential in enhancing our work’s quality.

We have released our code, and it is available at the link listed in the abstract.

In this common response, we would like to explain the figures newly uploaded:

**1. Figure A:** We display the weights of different layers after probing-based importance estimation. These weights can be seen as a "compression" of the cycle-consistency-based ranking process, directly integrated into the decoding process without the need for a RES model, thus enhancing the practicality of our method. The effectiveness of the important weights is validated in Table 3 of our paper.

**2. Figure B:** We have listed some prompt-response examples from the PHD benchmark to illustrate the quantification of hallucinations. We hope this can help answer the questions of reviewers MMWB and 85oQ.

**3. Figure C (1)-(5):** Visualization of multi-modal alignment in intermediate layers . We delve deeper into showcasing the transition of multi-modal alignment across different layers of a well-trained MLLM, as well as the potential impact of this transition process on the region-level understanding capabilities of intermediate layers. We can observe the following phenomena:
- The degree of multi-modal alignment varies across different layers. More specifically, in the early layers, the relative distance between visual tokens and language tokens is bigger than what in the later layers.
- The shift in language tokens across layers is greater than that of other types of tokens.
- The distance between the last language token (used for next token prediction) and region-related tokens does not change monotonically.

   Despite the limitations of the specific distance in measuring token alignment comprehensively, our observations suggest that the multi-modal alignment of intermediate layers of a well-trained MLLM undergoes a transitional phase, with these varied transitional states potentially providing better region understanding compared to the final layer. Reviewer MMWB and 85oQ might have interests in it.

**4. Figure C (6):** We provide a failure case according to the suggestion of reviewer 85oQ.

---

### Decision · Program_Chairs · 2024-09-25

**Decision:**

Accept (poster)

**Comment:**

This paper receives mostly positive rates (1 strong accept, 1 weak accept, 1 borderline accept and 1 borderline reject) thanks to the efficacy of the proposed model and convincing experimental results. Although overall reviews are positive, reviews still have some concerns, which should be addressed in the final version.